# Gemcitabine + Cisplatin + S-1 Treatment for Advanced Cholangiocarcinoma: Cost-Effective, with Better Progression-Free Survival Versus Standard Treatment with Gemcitabine + Cisplatin + Durvalumab

**DOI:** 10.3390/cancers17243971

**Published:** 2025-12-12

**Authors:** Yusuke Morita, Rie Sugimoto, Miho Kurokawa, Yuki Tanaka, Takeshi Senju, Toshimitsu Ichimaru, Lingaku Lee, Yusuke Niina, Terumasa Hisano, Masayuki Furukawa, Keishi Sugimachi, Masatake Tanaka

**Affiliations:** 1Department of Hepato-Biliary-Pancreatology, NHO Kyushu Cancer Center, Fukuoka 811-1395, Japan; morita.yusuke.238@m.kyushu-u.ac.jp (Y.M.); kurokawa.miho.zu@mail.hosp.go.jp (M.K.); tanaka.yuki.ep@mail.hosp.go.jp (Y.T.); senju.takeshi.mx@mail.hosp.go.jp (T.S.); ichimaru.toshimitsu.so@mail.hosp.go.jp (T.I.); lee.lingaku.wt@mail.hosp.go.jp (L.L.); niina.yusuke.uk@mail.hosp.go.jp (Y.N.); hisano.terumasa.ur@mail.hosp.go.jp (T.H.); furukawa.masayuki.kc@mail.hosp.go.jp (M.F.); 2Department of Medicine and Bioregulatory Science, Graduate School of Medical Sciences, Kyushu University, Fukuoka 812-8582, Japan; tanaka.masatake.656@m.kyushu-u.ac.jp; 3Department of Hepatobiliary and Pancreatic Surgery, NHO Kyushu Cancer Center, Fukuoka 811-1395, Japan; ksugimachi@gmail.com

**Keywords:** biliary tract cancer, systemic chemotherapy, immune checkpoint inhibitors, drug costs

## Abstract

Gemcitabine plus cisplatin combined with an immune checkpoint inhibitor is established as the standard treatment for advanced cholangiocarcinoma. Meanwhile, gemcitabine plus cisplatin plus S1 (GCS) has shown some efficacy compared with gemcitabine plus cisplatin alone in several countries; however, no studies have directly compared these two regimens. This study was performed to compare the efficacy, safety, and costs of GCS versus gemcitabine plus cisplatin plus durvalumab (GCD) using propensity score matching. Twenty-seven patients were selected for each group from 52 GCS cases and 44 GCD cases. Overall survival and objective response rates were comparable, while progression-free survival favoured GCS. Treatment-related costs were significantly lower with GCS. Taken together, these findings suggest that GCS may be considered as a potentially promising standard treatment option for advanced biliary tract cancer.

## 1. Introduction

Biliary tract cancer (BTC) includes malignant tumours such as intrahepatic cholangiocarcinoma, gallbladder cancer, extrahepatic cholangiocarcinoma and cancer of the papilla of Vater [1]. Currently, surgery remains the only curative treatment; however, early detection is often difficult and the prognosis is generally poor [2,3]. Furthermore, the postoperative recurrence rate remains high [4], underscoring the need for effective chemotherapy regimens [5]. Combination therapy with gemcitabine plus cisplatin (GC) has long been regarded as the standard chemotherapy, although its associated median survival of 11.7 months is unsatisfactory [6].

In 2022, Oh et al. demonstrated that combination therapy with GC plus durvalumab (GCD) improved overall survival (OS), with a hazard ratio (HR) of 0.8 in the TOPAZ-1 trial [7]. The median OS with this regimen in the final analysis was 12.9 months [8]. The KEYNOTE-966 trial also reported that GC plus pembrolizumab (GCP) improved median OS to 12.7 months, with an HR of 0.83 compared with GC [9]. Consequently, GC combined with immune checkpoint inhibitors (ICIs) is now positioned as the standard treatment for advanced cholangiocarcinoma [10].

Notably, Ioka et al. reported in the KHBO1401-MITSUBA trial that GC plus S-1 (GCS) improved median OS to 13.5 months compared with GC alone, with an HR of 0.79 [11]. Although GCS therapy is currently approved only in a limited number of countries, it combines oral treatment with intravenous cytotoxic agents. Potential advantages include reducing the frequency of intravenous outpatient visits and better control of drug costs. Although the non-inferiority of GS therapy to GC therapy has been reported by Morizane et al. [12], no randomised controlled trials directly comparing GCD and GCS therapies have been published.

In this study, we prospectively registered patients undergoing either GCS or GCD therapy to evaluate the similarity of these therapies for unresectable advanced BTC and to examine treatment costs. We assessed OS, PFS and objective response rates (ORRs). We then performed propensity score matching (PSM) to adjust for patient background factors and re-evaluated OS, PFS and ORR while also examining costs incurred until treatment completion.

## 2. Materials and Methods

### 2.1. Patients

All patients with advanced BTC at our hospital for whom radical surgery was deemed unfeasible (based on the decisions of our Hepatobiliary Pancreatic Surgery, Internal Medicine and Radiology Department conferences), or who experienced postoperative recurrence and received GCS or GCD as initial treatment between April 2020 and April 2024, were prospectively registered, provided that informed consent for participation in the study was obtained. We excluded patients for whom resection was considered potentially feasible at diagnosis and who received chemotherapy as neoadjuvant therapy with the intention of surgery. BTC comprised intrahepatic cholangiocarcinoma, extrahepatic cholangiocarcinoma, gallbladder cancer or histologically confirmed cancer of the ampulla of Vater. In principle, every effort was made to register all eligible patients. The choice of treatment was left to the judgement of the attending physician and the patient’s wishes.

The study was approved by the Ethics Committee of the Kyushu Cancer Center (approval number: 2020-14) and conformed to the principles of the 1975 Declaration of Helsinki.

### 2.2. Evaluation of Therapeutic Response and Safety

Before treatment, patients generally underwent a medical history review, physical examination, imaging with contrast-enhanced computed tomography or magnetic resonance imaging, and blood tests. Where clinically feasible, tumour biopsy was performed before treatment to establish a definitive pathological diagnosis. Physical examinations and blood tests were conducted before the start of each treatment course. Carcinoembryonic antigen and glycoantigen 19-9 concentrations were measured at study enrolment and monthly thereafter. Toxicity was assessed using the Common Terminology Criteria for Adverse Events version 4.0 [13]. In patients with measurable target lesions, the ORR was evaluated in accordance with the Response Evaluation Criteria in Solid Tumours (RECIST) version 1.1 [14]. Additional imaging studies were performed at the discretion of the treating physician.

### 2.3. Treatment Protocol

In the GCS group, gemcitabine and cisplatin were administered intravenously on Day 1 at doses of 1000 and 25 mg/m^2^, respectively, in 14-day cycles. Oral S-1 was given twice daily for 7 consecutive days starting on Day 1 of each cycle. The S-1 dose was calculated on the basis of body surface area (BSA) as follows: BSA < 1.25 m^2^, 80 mg/day; BSA ≥ 1.25 m^2^ and <1.5 m^2^, 100 mg/day; and BSA ≥ 1.5 m^2^, 120 mg/day. Chemotherapy was initiated on Day 1 and repeated when the following criteria were met: neutrophil count ≥ 1500/µL, platelet count ≥ 100,000/µL, total bilirubin ≤ 3.0 mg/dL, aspartate aminotransferase and alanine aminotransferase ≤ 150 IU/L and creatinine ≤ 1.2 mg/dL. GCS was administered only if there were no incidents of fever due to infection (≥38 °C), Grade ≥ 2 stomatitis or diarrhoea or Grade ≥ 3 non-haematological toxicities (excluding blood test abnormalities unrelated to the investigational drug). If these criteria were not met, chemotherapy was deferred until recovery. S-1 was discontinued during the treatment period if any of the following criteria were met: neutrophil count < 1000/mm^3^, platelet count < 75,000/µL, total bilirubin > 3.0 mg/dL, aspartate aminotransferase and alanine aminotransferase > 150 IU/L, or if any of the aforementioned precondition thresholds were exceeded. If gemcitabine-related Grade 4 neutropenia, Grade 4 thrombocytopenia, febrile neutropenia or Grade 3 non-haematological toxicity occurred, the subsequent gemcitabine dose was reduced to 800 mg/m^2^. If toxicity persisted after this reduction, the dose was further reduced to 600 mg/m^2^. When necessary, subsequent gemcitabine doses were reduced by 20%. If diarrhoea, stomatitis, anorexia, nausea or fatigue (Grade 3) associated with S-1 occurred, the S-1 dose in the next cycle was reduced as follows: 80/60, 100/80 or 120/100 mg/day (pre-dose/post-dose). If further dose reductions were required, the dose was adjusted in the next cycle to 60/50, 50/40, 40/20 or 20/0 mg/day (pre-dose/post-dose). Cisplatin was interrupted until recovery if the patient developed cisplatin-related neuropathy (Grade ≥ 2) or hearing impairment. Each agent could be re-escalated to the previous dose once toxicity was deemed sufficiently resolved. Protocol treatment was discontinued upon any of the following: deterioration in general condition due to disease progression, unacceptable or recurrent treatment-related toxicity, patient refusal or tumour response offering the possibility of curative resection.

In the GCD group, durvalumab (1500 mg) was administered intravenously on Day 1 of each 21-day cycle, while gemcitabine (1000 mg/m^2^) and cisplatin (25 mg/m^2^) were given on Days 1 and 8 of each cycle. After eight cycles, durvalumab 1500 mg monotherapy was continued until clinical or radiographic disease progression, unacceptable toxicity, patient refusal or fulfilment of other discontinuation criteria. Durvalumab was suspended until toxicity improved to Grade ≤ 1 for Grade 2 interstitial pneumonia, hepatic dysfunction, colitis, diarrhoea, renal impairment, neuropathy or skin disorders. Treatment was discontinued if any of these immune-related adverse events (IrAEs) progressed to Grade ≥ 3, or if Grade ≥ 2 myocarditis, myasthenia gravis or encephalitis occurred. For gemcitabine-related Grade 4 neutropenia, Grade 4 thrombocytopenia, febrile neutropenia or Grade 3 non-haematological toxicity, subsequent gemcitabine doses were reduced to 800 mg/m^2^. If toxicity persisted after this reduction, the dose was further reduced to 600 mg/m^2^. If additional reductions were necessary, subsequent doses were reduced by 20%. Cisplatin was interrupted until recovery if the patient developed cisplatin-related neuropathy (Grade ≥ 2) or hearing impairment.

### 2.4. Measured Parameters

We analysed the following patient data: clinical characteristics; treatment duration; medication costs until treatment completion; therapeutic response, namely OS, PFS, ORR and disease control rate (DCR); presence or absence of conversion therapy; and adverse events, including IrAEs. OS was defined as the time between the start date of GCS or GCD administration and the date of death. PFS was defined as the time between the start date of GCS or GCD administration and the date of the final follow-up examination, disease progression or death, whichever occurred first. Subsequently, PSM was performed between the GCS and GCD groups using age, sex, stage, local progression/distant metastasis, primary tumour site (gallbladder/others) and unresectable/postoperative recurrence as the evaluated factors (Figure 1). Using the 27 matched pairs extracted from the PSM, OS, PFS, ORR and DCR were compared between the GCS and GCD groups. Drug costs incurred until the end of treatment were also compared. Survival follow-up was conducted for patients who remained alive after treatment completion.

### 2.5. Statistical Analysis

Continuous data are presented as medians and ranges. The GCS (n = 52) and GCD (n = 44) groups were matched using PSM to reduce the influence of confounding factors. The propensity score was estimated using logistic regression with treatment assignment. Six factors considered to affect the prognosis of progressive bile duct cancer, given differences in patient backgrounds, were included as baseline variables: age, sex, stage, local progression/distant metastasis, primary tumour site (gallbladder/others) and unresectable/postoperative recurrence. The propensity score is the probability that a patient would be in either the GCS group or the GCD group based on these six factors. The propensity scores for the GCS and GCD groups were 0.633 ± 0.158 and 0.311 ± 0.161, respectively (mean ± standard deviation). The area under the receiver operating characteristic curve was 0.931. This propensity score was used for one-to-one nearest-neighbour matching, with the calliper width set at 0.15. This resulted in the selection of 27 participants from each of the GCS and GCD groups. The propensity scores after matching were 0.311 ± 0.161 and 0.611 ± 0.161 for the GCS and GCD groups, respectively. In the matched cohort, the standardised mean difference for stage was reduced from 2.14 to 0.28, and for disease extent from 0.40 to 0.09, indicating substantially improved balance for these key prognostic factors. Some residual imbalance remained for age, sex and gallbladder cancer; therefore, the results should be interpreted with caution. The covariate balance before and after matching is summarised in a Love Plot (Appendix A). The distribution of propensity scores before and after matching in the GCS and GCD groups is shown in Appendix A. OS and PFS after PSM are reported as median values, expressed in months, with 95% confidence intervals (CIs). Survival curves were plotted using the Kaplan–Meier product-limit method. The roles of other variables in survival were assessed using the log-rank test. ORR and DCR were analysed using Fisher’s exact probability test; *p* < 0.05 was considered statistically significant. All statistical analyses were performed using JMP Pro version 17.0 (SAS Institute Inc., Cary, NC, USA). Graphs were generated using GraphPad Prism (version 10.6.1; GraphPad Software, San Diego, CA, USA; https://www.graphpad.com/, accessed on 10 January 2025).

## 3. Results

### 3.1. Results Before PSM

#### 3.1.1. Patient Characteristics Before PSM

We enrolled patients at our hospital who were deemed ineligible for curative surgery (determined by our hepatobiliary-pancreatic multidisciplinary team) or who experienced postoperative recurrence, and who received GCS as initial treatment between April 2020 and April 2024. Of these 54 patients, 52 were enrolled after excluding two whose disease was considered resectable at diagnosis and who received preoperative adjuvant chemotherapy. Additionally, 44 patients were enrolled from the 45 who received GCD between March 2023 and April 2024, after excluding one patient from whom consent for study participation could not be obtained (Figure 1). BTC comprised intrahepatic cholangiocarcinoma, extrahepatic cholangiocarcinoma, gallbladder cancer or histologically confirmed cancer of the ampulla of Vater. All patients receiving GCS or GCD who met the inclusion criteria were enrolled, except for the aforementioned patient who did not provide consent. The patients’ background characteristics are shown in Table 1. The overall male-to-female ratio was 62.5% male. Although not statistically significant, the GCS group had a slightly higher proportion of men (71%) than did the GCD group (52%) (*p* = 0.0899). The median age was 66.5 years. The median age in the GCS group was 65 (31–84) years, whereas the GCD group had a significantly older median age of 68 (48–85) years (*p* = 0.0437). The primary sites were intrahepatic (38.5%), hilar (22.9%), distal bile duct (11.4%), gallbladder (25.0%) and ampullary (2.1%). Intergroup comparisons of primary site distribution revealed no significant differences (chi-square test, χ^2^ = 5.57; degrees of freedom = 4; *p* = 0.234). Regarding stage, the GCS group included patients with relatively early-stage disease (Stage I: 6%, Stage II: 15%, Stage III: 13%, Stage IV: 65%), whereas the GCD group contained no patients with early-stage disease (Stage III: 13%, Stage IV: 86%). There was a significant difference between groups, with the GCS group having more early-stage cases (*p* = 0.0002). Regarding primary unresectability or postoperative recurrence, only 4% of patients in the GCS group had postoperative recurrence, whereas 32% of the GCD group did, indicating a significant difference, with more patients having primary unresectable disease in the GCS group. When examining whether unresectable status was due to local progression or distant metastasis, the GCS group had rates of 38% local progression and 62% distant metastasis, whereas the GCD group had 20% local progression and 80% distant metastasis. Although not statistically significant, there was a tendency towards fewer distant metastases in the GCS group (*p* = 0.0555). The median observation period was 18.6 months for GCS and 12.2 months for GCD, with the longest observation period being 57.9 months for GCS and 25.2 months for GCD.

#### 3.1.2. Comparison of OS and PFS Before PSM

There was no significant difference in OS between the GCS and GCD groups (18.6 versus 12.2 months, respectively; *p* = 0.0935). PFS in the GCS group was significantly superior to that in the GCD group at 10.2 months (6.5–13.8 months) versus 6.2 months (3.2–8.8 months), respectively (*p* = 0.0053) (Figure 2a,b). Given the possibility that treatment efficacy may differ according to patient characteristics, subgroup analyses were conducted comparing both treatments by age (elderly patients aged ≥70 years versus non-elderly patients), primary tumour site, renal function (good versus poor), and the presence or absence of ascites (Appendix A). Treatment outcomes for GCD and GCS did not differ among younger patients, those with normal renal function, or those without ascites. However, in elderly patients and those with impaired renal function, GCS showed little deterioration in outcomes, whereas GCD was associated with a poorer prognosis. Because this study analysed a small number of cases, large-scale trials will be necessary in the future.

#### 3.1.3. Comparison of Treatment Efficacy Before PSM

Treatment efficacy was compared between the GCS and GCD groups. Both the response rate and the efficacy rate were significantly better in the GCS group (ORR: 36.5% versus 15.9%, respectively; *p* = 0.0211; DCR: 78.8% versus 52.2%, respectively; *p* = 0.0262). The number of patients receiving conversion therapy was also significantly higher in the GCS group than in the GCD group (28.8% versus 6.8%, respectively; *p* = 0.0040) (Table 2).

### 3.2. Analysis Results After PSM

#### 3.2.1. Patient Characteristics After PSM

To minimise differences in background factors, we performed PSM using the following variables: age, sex, stage, local progression/distant metastasis, gallbladder cancer/others and unresectable/postoperative recurrence. After propensity score matching (27 pairs, n = 54), most baseline characteristics were well balanced between the GCD and GCS groups. There were no significant differences in age (median 68 [62.5–73.0] versus 63 [48.0–73.0] years, *p* = 0.078), primary tumour site (*p* = 0.205), disease stage (*p* = 0.501), disease status (recurrent versus unresectable, *p* = 0.728), or extent of disease (locally advanced versus metastatic, *p* = 1.000). Sex showed a borderline imbalance (male 59.3% versus 29.6%, *p* = 0.054), but this difference did not reach statistical significance. The standardised mean differences before and after PSM are shown in Appendix A.

No significant differences in background factors remained after PSM (Table 3).

#### 3.2.2. Comparison of PFS and OS After PSM

There was no significant difference in OS between the GCS and GCD groups at 28.9 months (13.3–not reached) versus 13.0 months (7.3–23.1), respectively (*p* = 0.0734). By contrast, PFS in the GCS group was significantly superior to that in the GCD group at 9.3 months (7.4–27.1) versus 4.8 months (2.7–8.8), respectively (*p* = 0.0077) (Figure 3a,b).

#### 3.2.3. Comparison of Treatment Efficacy Following PSM

Treatment efficacy was compared between the GCS and GCD groups. No significant differences were observed between the groups for either the response rate or the efficacy rate (ORR: 29.6% versus 14.8%, respectively; *p* = 0.327; DCR: 63.0% versus 48.1%, respectively; *p* = 0.412). There were also no significant differences in the number of patients receiving conversion therapy (18.5% versus 7.4%, respectively; *p* = 0.424) (Table 4).

### 3.3. Adverse Events

Table 5 shows the adverse events that occurred during treatment. The incidence of any adverse event across all grades was 94.2% in the GCS group and 84.1% in the GCD group, with no significant difference in frequency between the groups. Grade 3 and 4 adverse events also showed no significant difference in frequency between the groups. However, a significant difference was observed in the nature of the adverse events. Haematological toxicity (leucopenia and neutropenia) occurred in 44.2% and 48.1% of patients in the GCS group, respectively, which was significantly higher than the rates of 22.7% and 15.9% in the GCD group (*p* = 0.0099 and *p* = 0.0322, respectively). Regarding non-haematological toxicity, constipation was significantly more frequent in the GCD group (56.8% versus **30.8%**, *p* = 0.0132), while fatigue/malaise was significantly more common in the GCS group (73.1%) than in the GCD group (36.4%) (*p* = 0.0004). Chronic kidney injury was observed in 34.1% of patients in the GCD group, significantly more than in the GCS group (9.6%) (*p* = 0.0049). No irreversible adverse events were observed in either group. Discontinuation because of adverse events, specifically haematopoietic suppression and fatigue/malaise, occurred in six patients (11%) in the GCS group. In the GCD group, three patients (6.8%) discontinued treatment: one due to IrAE nephritis and two due to cerebral infarction.

### 3.4. Drug Costs

S-1 is approved in Japan as a treatment for cholangiocarcinoma, with a drug price for the original product of Japanese yen (¥)297.5 for each 20 mg tablet and ¥367 for each 25 mg tablet. Calculated for a body surface area of 1.6 m^2^, the drug cost for one GCS cycle is approximately ¥22,750. Conversely, GCD costs approximately ¥1.65 million in Japan for the same body surface area. S-1 is not approved in the USA and is approved only for gastric cancer in Europe. On the basis of its price, 42 tablets (20 mg) cost approximately 217 Euros (€); therefore, assuming GCS is administered, one course would cost approximately €1800. In comparison, GCD costs approximately €5600 per course (per 21-day treatment cycle), although this varies by country. In the present study, we calculated the total costs incurred in Japan until treatment completion, including dose reductions. The costs for GCS versus GCD were €3473 (95% CI: 2233–4465) versus €83,619 (95% CI: 59,702–106,884) (US dollars $4038 (95% CI: 2596–5192) versus $97,231 (95% CI: 69,423–124,282)), respectively, demonstrating a significant difference, with GCS being less expensive (*p* = 0.0001). Even in PSM-matched cases, the cost of GCS versus GCD was €5461 (95% CI: €1978–€8428) versus €83,592 (95% CI: €59,684–€119,368) (US dollars $6350 (95% CI: 2300–9800) versus $97,200 (95% CI: 69,400–138,800)) by treatment end (*p* = 0.0001), indicating that GCS remains significantly less expensive (*p* = 0.0008). The costs analysed here relate solely to drug costs and do not include supportive care or toxicity-related medical costs.

## 4. Discussion

The incidence of cholangiocarcinoma is increasing worldwide [15,16], with most cases detected at locally advanced or metastatic stages, making systemic drug therapy the cornerstone of treatment [17,18]. The cornerstone of first-line systemic therapy has long been multi-agent combination treatment, with GC as the long-standing main component [6]. Compared with GC, the TOPAZ-1 trial demonstrated the superiority of GCD therapy [7,8], while KEYNOTE-966 showed the superiority of GCP therapy [9]. The KHBO1401-MITSUBA trial demonstrated the superiority of GCS over GC [11]. Whereas both GCD and GCP combine a cytotoxic agent with an ICI, GCS comprises three cytotoxic anticancer agents. To date, no randomised trial has directly compared GC plus an ICI with GCS. The present study is, to our knowledge, the first prospective observational study to demonstrate the similarity of GCS compared with GCD. Our findings showed that compared with GCD, GCS resulted in no significant difference in OS but demonstrated a significant prolongation in PFS. Regarding adverse events, although differences were observed in the frequency of certain events between the two regimens, no overall difference in the frequency or severity of adverse events was noted. In the GCD regimen, IrAEs were observed because of the inclusion of ICIs. Conversely, the GCS regimen, which does not contain ICIs and therefore does not induce IrAEs, includes three cytotoxic agents, resulting in a higher incidence of leucopenia and neutropenia. The overall frequency of adverse events and serious adverse events was broadly comparable between GCD and GCS. Furthermore, the adverse event profiles observed in the present study were generally consistent with those reported in the literature [7,11].

This study represents the first report, to our knowledge, comparing these two regimens; however, because it was an observational rather than a randomised study, there were variations in patient background factors. Therefore, we used PSM to adjust for these differences. The results remained similar after PSM, demonstrating the similarity of GCS compared with GCD. Previous reports demonstrated superiority of the GCD regimen over GC in terms of PFS (7.2 months versus 5.7 months; HR 0.75) and OS (12.8 months versus 11.5 months; HR 0.80) [7,8]. GCS also demonstrated superiority over GC for PFS (7.4 months versus 5.5 months; HR 0.75) and OS (13.5 months versus 12.6 months; HR 0.79) [11]. In the present study, no significant differences in OS after PSM were observed: OS after PSM was 13.0 months for GCD and 28.9 months for GCS, while PFS after PSM was 4.8 months for GCD and 9.3 months for GCS. In this study, no significant difference in OS was observed post-PSM: OS after PSM was 13.0 months in the GCD group and 28.9 months in the GCS group, both longer than in the MITSUBA trial. This may be attributable to several patients in the GCS group undergoing conversion surgery and achieving cancer-free status, potentially contributing to improved OS. Conversely, post-PSM PFS was 4.8 months in the GCD group and 9.3 months in the GCS group, with the GCD group showing a shorter PFS than in TOPAZ-1. This is likely related to the lower proportion of intrahepatic cholangiocarcinoma (48.1% versus 55.7% in TOPAZ-1), the greater number of locally advanced cases (26% versus 11%) and the shorter median observation period of 12.2 months in this study compared with TOPAZ-1. Both GCD and GCS regimens include gemcitabine and cisplatin but differ in drug formulation and dosing intervals. GCD is a 21-day cycle, where GC is administered on Day 1 with durvalumab added as an infusion, followed by GC on Day 8. From the ninth cycle onwards, durvalumab is administered as monotherapy. By contrast, GCS is a 14-day cycle involving GC on Day 1, with S-1 administered orally from Days 1–7.

Drug prices vary by country; however, when we calculated treatment costs in Japan, including dose reductions, the total drug costs incurred until treatment completion were €3473 (95% CI: 2233–4465) for GCS versus €83,619 (95% CI: 59,702–106,884) for GCD, demonstrating a significant difference in favour of GCS as the more cost-effective option (*p* = 0.0001). Even in cases matched using PSM, GCS versus GCD costs were €5461 versus €83,592 (US dollars $6350 versus $97,200) by treatment end, again demonstrating a significant cost advantage for GCS (*p* = 0.0008). The analysis here covers only drug costs and excludes costs related to therapeutic interventions, toxicity management, personnel and equipment. Even taking this into account, the cost difference is clear, and considering the similarity in PFS and OS, GCS is a possible option for standard treatment. In Western cohorts, albumin-bound paclitaxel plus GC [19] and mFOLFIRINOX [20] failed to improve prognosis compared with GC. GCS is approved only in Asia, including Japan, whereas S-1 is approved in Europe as a gastric cancer treatment. Because the pharmacokinetics and pharmacodynamics of S-1 differ between Asians and people of European descent, similar efficacy may not be expected in the latter population [21]. However, substituting drugs such as capecitabine may enable treatment at a lower cost [22,23]. There are multiple reports indicating that BTC frequently leads to severe complications and requires emergency hospitalisation, resulting in extremely high treatment costs [24,25]. Although this is database-based research, Chamberlain et al. reported that the average total monthly CCA-related healthcare expenditure per patient was $7743, with healthcare services ($6685) constituting a larger proportion of monthly costs than treatment costs ($1058) [26]. Furthermore, Healey et al. reported that the financial burden increased as treatment progressed from one course to two or three courses [24]. Considering these factors, the triple combination therapy of GCS agents, which is cost-effective, offers equivalent therapeutic efficacy and preserves the option of ICI, may be regarded as a viable treatment alternative.

This study has limitations. First, it was a prospective observational study rather than a randomised controlled trial. Additionally, treatment was chosen by the attending physician and the patient, which may have introduced bias in patient background factors. Although background factors were matched using PSM, it remains unclear whether completely equivalent backgrounds were achieved. Second, this was a single-centre study rather than a multicentre trial, resulting in a limited number of cases. Third, the observation period was short. Particularly for GCD, the longest follow-up period was 25.2 months, indicating inadequate tracking. Nevertheless, the present study demonstrated the similarity of GCS compared with GCD, which is currently the standard treatment worldwide, and represents a valuable dataset showing that equivalent therapeutic effects can be expected at a lower cost with GCS. A randomised trial comparing GCS and GCD is currently underway in Japan [27], and its results are awaited. Furthermore, in Europe and the USA, treatments using capecitabine and similar agents may offer equivalent efficacy. Considering the significant difference in cost-effectiveness, there is scope for prospective trials comparing GCS and GCD.

## 5. Conclusions

This prospective observational study using PSM demonstrated that GCS may be comparable to GCD as chemotherapy for advanced BTC. In this study, GCS was associated with superior PFS compared with GCD, equivalent OS and ORR, and similar adverse event rates. Furthermore, GCS may be significantly less expensive than GCD.

## Figures and Tables

**Figure 1 cancers-17-03971-f001:**
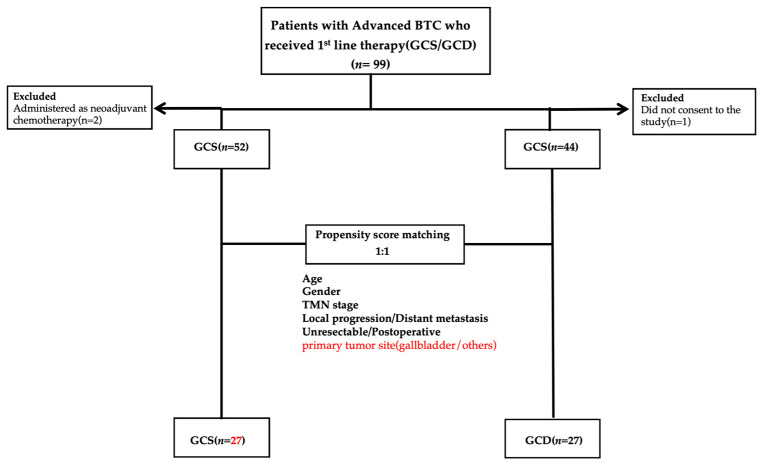
Study design. In total, 99 patients with BTC treated with GCD or GCS as first-line systemic treatment were evaluated. During the study, three patients were excluded, resulting in 96 patients with BTC. Propensity score matching was then performed on the data of these 96 patients, resulting in 54 patients included in the evaluation. Abbreviations: BTC: biliary tract cancer, GCD: gemcitabine plus cisplatin plus durvalumab, GCS: gemcitabine plus cisplatin plus S-1.

**Figure 2 cancers-17-03971-f002:**
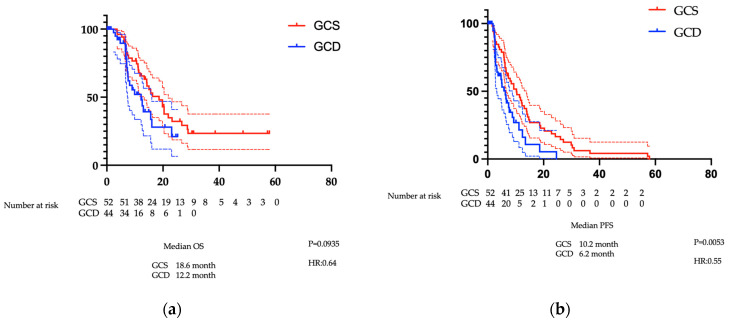
Survival analysis for the GCS and GCD groups for all cases of biliary tract cancer: (**a**) OS (months), (**b**) PFS (months). The red lines indicate GCS data, and the blue lines indicate GCD data. Each dotted line indicates the 95% confidence interval. Abbreviations: GCS: gemcitabine plus cisplatin plus S-1, GCD: gemcitabine plus cisplatin plus durvalumab, OS: overall survival, PFS: progression-free survival.

**Figure 3 cancers-17-03971-f003:**
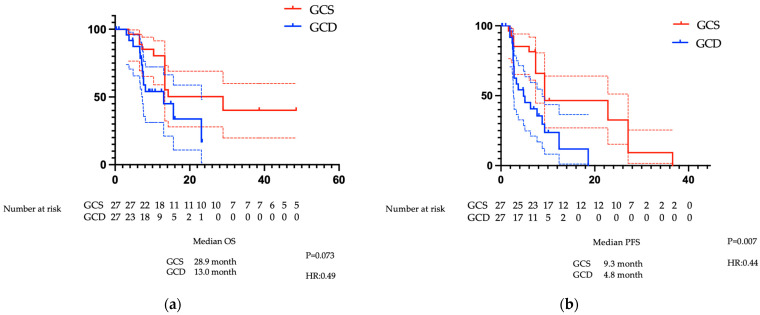
Analysis of survival periods for GCS and GCD after PSM in cases of biliary tract cancer: (**a**) OS (months), (**b**) PS (months). The red lines indicate GCS data, and the blue lines indicate GCD data. Each dotted line indicates the 95% confidence interval. Abbreviations: GCS: gemcitabine plus cisplatin plus S-1, GCD: gemcitabine plus cisplatin plus durvalumab, OS: overall survival, PFS: progression-free survival, PSM: propensity score matching.

**Table 1 cancers-17-03971-t001:** Patient characteristics before PSM.

Characteristic	All Patients (n = 96)	GCS (n = 52)	GCD (n = 44)	*p*
Gender (male %)	60 (62.5)	37 (71)	23 (52)	0.0899
Age, years	66.5 (31–85)	65 (31–84)	68 (42–85)	0.0437 *
Performance Status (0/1)	91 (95%)/5 (5%)	51 (98%)/1 (2%)	40 (90%)/4 (9%)	0.176
Primary tumour site				0.234
Intrahepatic bile duct	37 (38.5)	20 (38.5)	17 (38.6)	
Perihilar	22 (22.9)	14 (26.9)	8 (18.2)	
Distal	11 (11.4)	7 (13.4)	4 (9.1)	
Gall bladder	24 (25.0)	9 (17.3)	15 (34.1)	
Ampullary	2 (2.1)	2 (3.9)	0 (0)	
Stage I/II/III/IV	3/8/13/72	3/8/7/34	0/0/6/38	0.0134 *
	(3/8/13/75)	(6/15/13/65)	(0/0/13/86)	
Recurrent	16 (17)	2 (4)	14 (32)	0.0002 *
Unresectable	80 (83)	50 (96)	30 (68)	
Locally advanced	29 (30)	20 (38)	9 (20)	0.0555
Metastatic	67 (70)	32 (62)	35 (80)	
Ascites (Yes/No)	12/84	4 (8)/48 (92)	8 (18)/36 (82)	0.13
eGFR (mL/min/1.73 m^2^)	72 (55–127)	73 (66.5–89.3)	69.5 (59.5–82.0)	0.12
mALBI (G1/2a/2b/3)	30/20/33/45	13/11/20/8	17/9/11/37	0.43
CEA (ng/mL)	3.7	3.3 (2.35–6.5)	3.65 (2.0–14.7)	0.71
CA19-9 (U/mL)	301.5 (12.2–2332)	197 (21.2–2382)	349 (12.24–1217)	0.87
Complications (Diabetes, liver disease) (Yes/No)	19 (20)/77 (80)	13 (25)/39 (75)	6 (14)/38 (86)	0.24
total number of chemotherapy cycles		7 (2–22)	14 (2–32)	0.31

Data are expressed as median (range) or number (%). * Significant difference. Abbreviations: PSM: propensity score matching, GCS: gemcitabine plus cisplatin plus S-1, GCD: gemcitabine plus cisplatin plus durvalumab, eGFR: estimated Glomerular Filtration Rate, mALBI: modified Albumin-Bilirubin grade.

**Table 2 cancers-17-03971-t002:** Best tumour response by RECIST 1.1 before PSM.

	GCS (n = 52)	GCD (n = 44)
Complete response	1 (1.9)	0
Partial response	18 (34.6)	7 (15.9)
Stable disease	22 (42.3)	16 (36.3)
Progressive disease	8 (15.4)	15 (34.0)
Not evaluable	3 (5.8)	6 (13.6)

Abbreviations: RECIST: Response Evaluation Criteria in Solid Tumours, PSM: propensity score matching, GCS: gemcitabine plus cisplatin plus S-1, GCD: gemcitabine plus cisplatin plus durvalumab.

**Table 3 cancers-17-03971-t003:** Patient characteristics after PSM.

Characteristic	All Patients (n = 54)	GCS (n = 27)	GCD (n = 27)	*p*
Gender (male %)	24 (44.4)	8 (29.5)	16 (59.3)	0.054
Age, years	63 (48–73)	68 (48–73)	65.5 (62.5–73)	0.078
Primary tumour site				0.205
Intrahepatic bile duct	21 (38.8)	8 (29.6)	13 (48.1)	
Perihilar	10 (18.5)	4 (14.8)	6 (22.2)	
Distal	3 (5.5)	1 (3.7)	2 (7.4)	
Gall bladder	18 (33.3)	12 (44.4)	6 (22.2)	
Ampullary	2 (3.7)	2 (7.4)	0 (0)	
Stage I/II/III/IV	0/0/11/43 (0/0/20/80)	0/0/7/20 (0/0/26/74)	0/0/4/23 (0/0/15/85)	0.501
Recurrent	4 (8)	6 (22)	4 (15)	0.728
Unresectable	48 (92)	21 (78)	23 (85)	
Locally advanced	13 (25)	6 (22)	7 (26)	1.000
Metastatic	41 (75)	21 (78)	20 (74)	

Data are expressed as median (range) or number (%). Abbreviations: PSM: propensity score matching, GCS: gemcitabine plus cisplatin plus S-1, GCD: gemcitabine plus cisplatin plus durvalumab.

**Table 4 cancers-17-03971-t004:** Best tumour response by RECIST 1.1 after PSM.

	GCS (n = 27)	GCD (n = 27)
Complete response	0	0
Partial response	8 (29.6)	4 (14.8)
Stable disease	9 (33.3)	9 (33.3)
Progressive disease	4 (14.8)	12 (44.4)
Not evaluable	6 (22.2)	2 (7.4)
Conversion surgery	5 (18.5)	2 (7.4)

Note: Data are number (%). Abbreviations: RECIST: Response Evaluation Criteria in Solid Tumours, PSM: propensity score matching, GCS: gemcitabine plus cisplatin plus S-1, GCD: gemcitabine plus cisplatin plus durvalumab.

**Table 5 cancers-17-03971-t005:** Adverse events in all patients.

	GCS (n = 52)		GCD (n = 44)		*p*	
	All grade (%)	Grade3, 4 (%)	All grade (%)	Grade3, 4 (%)	All grade	Grade 3, 4
Any AE	37 (84.1)	24 (54.5)	49 (94.2)	29 (55.7)	0.1783	1.0000
Haematopoietic toxicity						
Anaemia	23 (52.3)	11 (25.0)	37 (71.2)	14 (26.9)	0.0899	1.0000
Leukopenia	12 (27.3)	10 (22.7)	**23 (44.2)**	19 (36.5)	0.0099 *	0.0998
Neutropenia	10 (22.7)	7 (15.9)	**25 (48.1)**	16 (30.8)	0.0322 *	0.1823
Thrombocytopenia	8 (18.2)	5 (11.4)	18 (34.6)	9 (17.3)	0.1060	0.5638
Febrile neutropenia	0	0	1 (1.9)	1 (1.9)	1.0000	1.0000
Non-haematopoietic toxicity						
Constipation	**25 (56.8)**	0	16 (30.8)	0	0.0132 *	
Loss of appetite	14 (31.8)	0	25 (48.1)	0	0.1445	
Nausea	7 (15.9)	0	16 (30.8)	0	0.0998	
Taste disturbance	7 (15.9)	0	13 (25.1)	0	0.3206	
Chronic kidney disease	**15 (34.1)**	1 (2.3)	5 (9.6)	1 (1.9)	0.0049 *	1.0000
Fatigue/malaise	16 (36.4)	1 (2.3)	**38 (73.1)**	1 (1.9)	0.0004 *	1.0000
Hand-Foot Syndrome	0	0	**8 (15.4)**	0	0.0070 *	
Oral mucositis	3 (6.8)	0	8 (15.4)	0	0.2177	
Diarrhoea	0	0	4 (7.7)	0	0.1224	
Thrombosis	3 (6.8)	0	2 (3.8)	0	0.6580	
Stroke	3 (6.8)	2 (4.5)	0	0	0.0927	0.2075
Hypertension	2 (4.5)	1 (2.3)	0	0	0.2075	0.4583
Peripheral neuropathy	1 (2.3)	0	6 (11.5)	0	0.1204	
Rotatory vertigo	1 (2.3)	1 (2.3)	0	0	0.4583	0.4583
Hypokalaemia	1 (2.3)	1 (2.3)	0	0	0.4583	0.4583
Heart failure	1 (2.3)	1 (2.3)	0	0	0.4583	0.4583
Oedema	1 (2.3)	0	0	0	0.4583	
Hypoglycaemia	1 (2.3)	0	0	0	0.4583	
Hiccups	1 (2.3)	0	0	0	0.4583	
Hyponatraemia	0	0	2 (3.8)	1 (1.9)	0.4982	1.0000
Anaphylactic shock	0	0	1 (1.9)	1 (1.9)	1.0000	1.0000
IrAE						
Skin disorders	**6 (13.6)**	1 (2.3)	0	0	0.0076 *	0.4583
IrAE nephritis	1 (2.3)	1 (2.3)	0	0	0.4583	0.4583
Pancreatitis	1 (2.3)	1 (2.3)	0	0	0.4583	0.4583
Elevated AST	1 (2.3)	1 (2.3)	0	0	0.4583	0.4583
Elevated ALT	1 (2.3)	1 (2.3)	0	0	0.4583	0.4583
Increased amylase	1 (2.3)	0	0	0	0.4583	
Hyperthyroidism	1 (2.3)	0	0	0	0.4583	

Data are expressed as median (range) or number (%). Bold; indicates adverse events that were significantly more frequent. * Significant difference. Abbreviations: GCS: gemcitabine plus cisplatin plus S-1, GCD: gemcitabine plus cisplatin plus durvalumab, AE: adverse event, IrAE: immune-related adverse event, ALT: alanine aminotransferase, AST: aspartate aminotransferase.

## Data Availability

The raw data supporting the conclusions of this article will be made available by the authors upon request.

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
