# Peer review of "Gemcitabine + Cisplatin + S-1 Treatment for Advanced Cholangiocarcinoma: Cost-Effective, with Better Progression-Free Survival Versus Standard Treatment with Gemcitabine + Cisplatin + Durvalumab"

_cancers, 2025, doi:10.3390/cancers17243971_

Round 1

Reviewer 1 Report

Comments and Suggestions for Authors
  • Summary: This study by Morita et al., compares GCS versus the current GCD regimen for advanced cholangiocarcinoma in a prospective observational cohort with propensity score matching. The authors report similar OS between the two groups, but longer PFS, higher ORR, and dramatically lower drug costs with GCS. The topic is relevant, especially given the financial burden of immunotherapy. However, several key details are missing, and the manuscript needs clarification before the results can be interpreted reliably. No additional experiments are required.
  • Specific comments:
  1. ORR values and their interpretation are inconsistent between the abstract and the Results/Tables. These numbers need correction and should be presented clearly for both unmatched and PSM cohorts.
  2. Use of the term “noninferior” is not appropriate for this observational study and should be replaced with “similar” or “comparable.”
  3. The AE discussion needs slight clarification, since specific toxicities differ significantly between groups (Table 5). The statement “no overall difference” should be nuanced or supported more explicitly.
  4. OS outcomes in the GCD cohort are notably shorter than those reported in TOPAZ-1. A brief explanation of possible reasons (patient characteristics, follow-up length) would help contextualize this.
  5. Cost analysis should specify that it reflects drug cost only and does not include supportive care or toxicity-related medical costs.
  6. Conclusions should be phrased more cautiously, given the single-center observational design, small effective sample size after PSM, and relatively short follow-up in the GCD arm.
  7. No additional experiments are needed; these are primarily reporting and clarity issues.

Author Response

  • Thank you very much for taking the time to review this manuscript. Please find the detailed responses below and the corresponding revisions/corrections highlighted/in track changes in the re-submitted files
  • Specific comments:

1.ORR values and their interpretation are inconsistent between the abstract and the Results/Tables. These numbers need correction and should be presented clearly for both unmatched and PSM cohorts.

Thank you for pointing this out. This was an error in our abstract. To prevent confusion, we have reorganised the order of the description and made the correction (Abstract, lines 49–53).

2.Use of the term “noninferior” is not appropriate for this observational study and should be replaced with “similar” or “comparable.

Thank you for your precise observation. We have rephrased the term as ‘similar’ (lines 34, 386, 400, 431,455 and 463).

3.The AE discussion needs slight clarification, since specific toxicities differ significantly between groups (Table 5). The statement “no overall difference” should be nuanced or supported more explicitly.

Thank you very much. We agree with your point. We have added clarification noting that although the overall frequency of adverse events did not differ between groups, the specific toxicities did show significant differences (lines 336, 344–348).

4.OS outcomes in the GCD cohort are notably shorter than those reported in TOPAZ-1. A brief explanation of possible reasons (patient characteristics, follow-up length) would help contextualize this.

Thank you very much for your invaluable advice. As you noted, OS in the GCD cohort after PSM was shorter than expected. After incorporating additional matching factors, as recommended by other reviewers, this difference became less pronounced. Nevertheless, PFS remained shorter, and we have added an explanation of the possible reasons for this (lines 407–416).

5.Cost analysis should specify that it reflects drug cost only and does not include supportive care or toxicity-related medical costs.

Thank you for your comment. We have clarified in the Discussion and Conclusions that the cost analysis reflects only drug costs and does not include supportive care or toxicity-related medical costs (lines 371–373 and 428–430).

6.Conclusions should be phrased more cautiously, given the single-center observational design, small effective sample size after PSM, and relatively short follow-up in the GCD arm.

We concur with your observation. We have revised the conclusion to adopt more cautious phrasing (lines 463–466 and 33–36).

7.No additional experiments are needed; these are primarily reporting and clarity issues.

Thank you very much for your peer review.

Reviewer 2 Report

Comments and Suggestions for Authors

This paper presents a prospective observational study of advanced cholangiocarcinoma using 
propensity score matching. The Authors reported the prognosis based on OS, PFS, ORR and costs of two treatments: GCS vs. GCD.

The paper is well-writen and the statistical analysis was proper designed and conducted.
I have only few remarks:

1. Page 1, Abstract, in the sentances '(6.3, 95% CI: 2.8–9.3 versus 11.4, 95% CI: 6.2–15.0; p=0.0142) and ORR (14.8% versus 40.7%; 
p=0.0309) for GCS versus GCD, respectively. The cost of GCD versus GCS was 113,472 ± 
72,198 versus 4622 ± 3109 US dollars by treatment end (p=0.0001). Conclusions: GCS was 
significantly less expensive than GCD)'  by mistake swapped GCS vs. GCD. The same on page 11 ' GCS versus GCD were €3961 ± 2664 versus €97,270 ± 61,873 (US dollars $4622 ± 3109 
versus $ 113,472 ± 72,198), respectively, demonstrating a significant difference, with GCS 
being less expensive (p=0.0001)'

2. Page 5, Statistical Analysis, 
For more clearity it is neccessary added that the propensive score is the probability that a patient would be in either the GCS group or the GCD group,
 based on five factors: age, gender, stage, local progression/distant metastasis, and unresectable/postoperative recurrence.

3. Page 5, Statistical Analysis, How was the propensity score estimated from the data? Was logistic regression used?

4. The size of the sample after PMS is 26 or 27? On page 1 of the abstract the sample size is 27, but on page 2 of the graphical abstract it is 26.
On the following pages there are also alternately 26 or 27.

5. What statistical program was used for the analyses and please provide the version of this program.

Author Response

This paper presents a prospective observational study of advanced cholangiocarcinoma using 
propensity score matching. The Authors reported the prognosis based on OS, PFS, ORR and costs of two treatments: GCS vs. GCD.

The paper is well-writen and the statistical analysis was proper designed and conducted.
I have only few remarks:

  1. Page 1, Abstract, in the sentances '(6.3, 95% CI: 2.8–9.3 versus 11.4, 95% CI: 6.2–15.0; p=0.0142) and ORR (14.8% versus 40.7%; p=0.0309) for GCS versus GCD, respectively. The cost of GCD versus GCS was 113,472 ± 72,198 versus 4622 ± 3109 US dollars by treatment end (p=0.0001). Conclusions: GCS was significantly less expensive than GCD)' by mistake swapped GCS vs. GCD. The same on page 11 ' GCS versus GCD were €3961 ± 2664 versus €97,270 ± 61,873 (US dollars $4622 ± 3109 versus $ 113,472 ± 72,198), respectively, demonstrating a significant difference, with GCS being less expensive (p=0.0001)'

Thank you for pointing this out. This was an error on our part. We have corrected it and have also standardised the order of presentation for GCS and GCD to improve clarity (lines 49–53).

  1. Page 5, Statistical Analysis, For more clearity it is neccessary added that the propensive score is the probability that a patient would be in either the GCS group or the GCD group, based on five factors: age, gender, stage, local progression/distant metastasis, and unresectable/postoperative recurrence.

Thank you for your advice. To clarify, the propensity score represents the probability of a patient being assigned to either the GCS group or the GCD group based on six factors: age, sex, disease stage, local progression/distant metastasis, primary tumour site (gallbladder/others), and unresectable status/postoperative recurrence (lines 196–197).

  1. Page 5, Statistical Analysis, How was the propensity score estimated from the data? Was logistic regression used?

Thank you for pointing this out. The propensity scores were estimated using logistic regression analysis, and we have added this information (lines 191–192).

  1. The size of the sample after PMS is 26 or 27? On page 1 of the abstract the sample size is 27, but on page 2 of the graphical abstract it is 26. On the following pages there are also alternately 26 or 27.

Thank you for pointing this out. Initially, the correct number was 26, but after incorporating additional matching factors as recommended by another reviewer, the PSM was recalculated and resulted in 27 matched pairs. We have therefore standardised the final number as 27 throughout the manuscript.

  1. What statistical program was used for the analyses and please provide the version of this program.

Thank you for your comment. We have specified the statistical programmes and their versions used for the analysis (lines 214–216).

Reviewer 3 Report

Comments and Suggestions for Authors

The manuscript is a strong observational study, with valuable insights into the cost-effectiveness of GCS compared to GCD for treating advanced BTC. After thoroughly reviewing the manuscript, I have provided some feedback and suggestions for improvement before it proceeds to publication.

Major comments
1. Kaplan-Meier survival curves are a good way to present OS and PFS, but including hazard ratios (HRs) with confidence intervals (CIs) would provide a better understanding of the relative risks between the groups (Graphic abstract, Figure 2 and 3).

2. The use of PSM is well-explained, but it would be beneficial to include more details about how well the matching process worked (e.g., balance of corvariates post-matching). Although no significant differences were found in background factors, the proportion of GB cancer can be appeared to be higher in the GCD group (38.5% vs 15.3%)

3. In figure 1 and graphic abstract, BCLC stage was decribed as one of the five factors for PSM matching. However, BCLC is the staging systetm for hepatocellular carcinoma (HCC), and AJCC or TNM staging system was more commonly used for BTC staging. Clarifying this distinction may be helpful for readers.

4. One-third of the GCD group’s best response was not evaluated (31.8%, Table 2), and this still persists after PSM matching (Table 3). Providing an explanation for why this discrepancy exists would be helpful.

5. The manuscript includes limited data on patient characteristics such as medical history, performance status, laboratory findings, total number of chemotherapy cycles, and cumulative chemotherapy doses. Including these details would provide a fuller understanding of the patient population and treatment regimen.

Minor comments
1. The subtitle of Table 4 should be revised from "before PSM" to "after PSM" for consistency.

2. The discussion section would benefit from more comparisons with existing studies beyond the trials mentioned, particularly those focusing on treatment costs, outcomes, and side effects in similar populations.

3. If possible, consider adding supplementary material that includes more detailed data or methodology, such as full PSM results or additional survival curves based on subgroup analyses (e.g., age, disease stage).

4. In Table 5, febrile neutropenia is incorrectly categorized under non-hematopoietic toxicities, and its incidence is too low (only 1.9% in the GCD group). It would be helpful to also report the proportion of adverse events (AEs) leading to chemotherapy discontinuation or death for a more comprehensive understanding.

Author Response

The manuscript is a strong observational study, with valuable insights into the cost-effectiveness of GCS compared to GCD for treating advanced BTC. After thoroughly reviewing the manuscript, I have provided some feedback and suggestions for improvement before it proceeds to publication.

Major comments
1. Kaplan-Meier survival curves are a good way to present OS and PFS, but including hazard ratios (HRs) with confidence intervals (CIs) would provide a better understanding of the relative risks between the groups (Graphic abstract, Figure 2 and 3).

Thank you for your valuable advice. We have added the HRs with 95% CIs to the graphical abstract and to Figure 2 and 3.

2. The use of PSM is well-explained, but it would be beneficial to include more details about how well the matching process worked (e.g., balance of corvariates post-matching). Although no significant differences were found in background factors, the proportion of GB cancer can be appeared to be higher in the GCD group (38.5% vs 15.3%)

Thank you for your extremely important observation. Indeed, as you pointed out, although there was no significant difference in the incidence of gallbladder cancer, there was a prognostic gap that could not be overlooked. Therefore, we re-ran the PSM with gallbladder cancer included as a new factor. As a result, the difference in ORR was no longer observed after PSM, while the significance of PFS remained unchanged. (line195, table 3)

The statistical analysis is as follows: In response to your comment, we reconstructed the propensity score model by including gallbladder cancer as an additional covariate. (line191-210). After incorporating this variable, the mean propensity scores were 0.633±0.158 for the GCD group and 0.311±0.161 for the GCS group. The updated propensity score model showed excellent discrimination between treatment groups, with an AUC of 0.931. These results have been added to the revised manuscript. In the matched cohort, the standardised mean difference for stage was reduced from 2.14 to 0.28 and for disease extent from 0.40 to 0.09, indicating substantially improved balance for these key prognostic factors. Some residual imbalance remained for age, sex and gallbladder cancer; therefore, the results should be interpreted with caution. The covariate balance before and after matching is summarised in a Love plot (Supplementary Figures 1 and 2). Furthermore, the standardised mean difference before and after PSM is also shown in Supplementary Table 1.

  1. In figure 1 and graphic abstract, BCLC stage was decribed as one of the five factors for PSM matching. However, BCLC is the staging systetm for hepatocellular carcinoma (HCC), and AJCC or TNM staging system was more commonly used for BTC staging. Clarifying this distinction may be helpful for readers.

Thank you for pointing that out. This was an error on our part. We used the TNM staging system, and we have corrected the figures accordingly. (Figure 1, graphic abstract,)

  1. One-third of the GCD group’s best response was not evaluated (31.8%, Table 2), and this still persists after PSM matching (Table 3). Providing an explanation for why this discrepancy exists would be helpful.

Thank you for your very astute observation. In the initial manuscript, the absence of evaluation in the GCD group was noted in eight patients (18.2%). Even this number was considered excessive, so we collected additional data—including by contacting the receiving hospitals—to reduce the number of patients with missing data. (Table 2, and3)

  1. The manuscript includes limited data on patient characteristics such as medical history, performance status, laboratory findings, total number of chemotherapy cycles, and cumulative chemotherapy doses. Including these details would provide a fuller understanding of the patient population and treatment regimen.

Thank you very much for your invaluable feedback. We have now added more detailed patient data to Table 1.

Minor comments
1. The subtitle of Table 4 should be revised from "before PSM" to "after PSM" for consistency.

Thank you for pointing that out. We have amended the subtitle to ‘After’.(Table4)

  1. The discussion section would benefit from more comparisons with existing studies beyond the trials mentioned, particularly those focusing on treatment costs, outcomes, and side effects in similar populations.

Thank you for your valuable advice. We have now incorporated additional citations to existing research, particularly with regard to treatment costs (lines 437–446).

  1. If possible, consider adding supplementary material that includes more detailed data or methodology, such as full PSM results or additional survival curves based on subgroup analyses (e.g., age, disease stage).

Thank you for your invaluable comments. We have provided detailed PSM results, including standardised mean differences, as supplementary material. In addition, we have conducted subgroup analyses comparing treatments by age group, primary tumour site, renal function and the presence of ascites, and have added these as supplementary material. (Supplementary Figure S1. S2, S3, S4, SupplementaryTable)

  1. In Table 5, febrile neutropenia is incorrectly categorized under non-hematopoietic toxicities, and its incidence is too low (only 1.9% in the GCD group). It would be helpful to also report the proportion of adverse events (AEs) leading to chemotherapy discontinuation or death for a more comprehensive understanding.

Thank you for your invaluable advice. We have reclassified febrile neutropenia as haematotoxicity. We re-examined the incidence rate in detail and confirmed that occurrences were indeed minimal during this period, which may be attributable to the relatively early adjustment of anticancer drug dosages. Furthermore, we have not identified any adverse events leading to death at this stage. The cases resulting in AE-related treatment discontinuation are described in lines 344–348.

Round 2

Reviewer 1 Report

Comments and Suggestions for Authors

I recommend acceptance of the manuscript. I thank the authors for thoroughly addressing all of my previous concerns.

Reviewer 2 Report

Comments and Suggestions for Authors

I accept this version of the manuscript. All my remarks are corrected.